# Effects of Sub-Minimum Inhibitory Concentrations of Imipenem and Colistin on Expression of Biofilm-Specific Antibiotic Resistance and Virulence Genes in *Acinetobacter baumannii* Sequence Type 1894

**DOI:** 10.3390/ijms232012705

**Published:** 2022-10-21

**Authors:** Abebe Mekuria Shenkutie, Jiaying Zhang, Mianzhi Yao, Daniel Asrat, Franklin W. N. Chow, Polly H. M. Leung

**Affiliations:** 1Department of Health Technology and Informatics, The Hong Kong Polytechnic University, Hong Kong SAR, China; 2Department of Microbiology, Immunology, and Parasitology, St. Paul’s Hospital Millennium Medical College, Addis Ababa P.O. Box 1271, Ethiopia; 3Department of Microbiology, Immunology, and Parasitology, School of Medicine, College of Health Sciences, Addis Ababa University, Addis Ababa P.O. Box 9086, Ethiopia

**Keywords:** *Acinetobacter baumannii*, biofilm, colistin, imipenem, antibiotic resistance, RNA sequencing, small RNA, virulence

## Abstract

Antibiotics at suboptimal doses promote biofilm formation and the development of antibiotic resistance. The underlying molecular mechanisms, however, were not investigated. Here, we report the effects of sub-minimum inhibitory concentrations (sub-MICs) of imipenem and colistin on genes associated with biofilm formation and biofilm-specific antibiotic resistance in a multidrug-tolerant clinical strain of *Acinetobacter baumannii* Sequence Type (ST) 1894. Comparative transcriptome analysis was performed in untreated biofilm and biofilm treated with sub-MIC doses of imipenem and colistin. RNA sequencing data showed that 78 and 285 genes were differentially expressed in imipenem and colistin-treated biofilm cells, respectively. Among the differentially expressed genes (DEGs), 48 and 197 genes were upregulated exclusively in imipenem and colistin-treated biofilm cells, respectively. The upregulated genes included those encoding matrix synthesis (*pgaB*), multidrug efflux pump (novel00738), fimbrial proteins, and homoserine lactone synthase (AbaI). Upregulation of biofilm-associated genes might enhance biofilm formation when treated with sub-MICs of antibiotics. The downregulated genes include those encoding DNA gyrase (novel00171), 30S ribosomal protein S20 (novel00584), and ribosome releasing factor (RRF) were downregulated when the biofilm cells were treated with imipenem and colistin. Downregulation of these genes affects protein synthesis, which in turn slows down cell metabolism and makes biofilm cells more tolerant to antibiotics. In this investigation, we also found that 5 of 138 small RNAs (sRNAs) were differentially expressed in biofilm regardless of antibiotic treatment or not. Of these, sRNA00203 showed the highest expression levels in biofilm. sRNAs regulate gene expression and are associated with biofilm formation, which may in turn affect the expression of biofilm-specific antibiotic resistance. In summary, when biofilm cells were exposed to sub-MIC doses of colistin and imipenem, coordinated gene responses result in increased biofilm production, multidrug efflux pump expression, and the slowdown of metabolism, which leads to drug tolerance in biofilm. Targeting antibiotic-induced or repressed biofilm-specific genes represents a new strategy for the development of innovative and effective treatments for biofilm-associated infections caused by *A. baumannii*.

## 1. Introduction

*Acinetobacter baumannii* is an emerging global antibiotic-resistant gram-negative bacteria that primarily causes biofilm-associated infections such as ventilator-associated pneumonia and catheter-related infection, both of which are resistant to conventional antibiotics [1,2,3].The capacity of *A. baumannii* to form biofilms enhances its survival in adverse environments, making it a successful nosocomial pathogen [4,5,6,7,8]. Recently, several environmental reservoirs in hospital settings were reported to be the primary sources for outbreaks of multidrug resistance (MDR) *A. baumannii* [9,10]. Biofilm formation protects *A. baumannii* from antibacterial agents and allows the pathogen to survive on abiotic and biotic surfaces [11]. In our previous study, we identified a strong biofilm-former, *A. baumannii* ST1894 [12]. The strain was susceptible to imipenem and colistin in the planktonic state but was highly resistant in the biofilm state, and the minimum inhibitory concentrations (MICs) of imipenem and colistin were 2048 and 32 times higher than those required to eradicate the planktonic cells [12]. Imipenem and colistin are the most common antibiotics used to treat multidrug-resistant *A. baumannii* strains; a decrease in antibiotic susceptibility in biofilm creates a significant problem in the control of *A. baumannii* infections in clinical settings [13,14]. Moreover, the decrease in susceptibility renders the antibiotic dosage sub-optimal, studies have shown that exposure to sub-optimal antibiotic dosages triggered biofilm formation and expression of antibiotic resistance genes [15,16]. Thus, exposure of *A. baumannii* biofilm to sub-MICs of imipenem and colistin may trigger transcriptional and post-transcriptional changes in the biofilm cells, which might result in biofilm-specific antibiotic resistance in *A. baumannii*.

Despite the fact that biofilm formation in *A. baumannii* decreases susceptibility to imipenem and colistin, little is known about the molecular mechanisms involved in the expression of antibiotic resistance and virulence when *A. baumannii* biofilm is exposed to sub-optimal doses of the two antibiotics. Here, we have conducted a comparative transcriptome study to determine the transcriptional changes when biofilm cells were treated with sub-MICs of imipenem and colistin. This study lays the foundation for future research on the development of novel therapeutics for treatment of biofilm-associated infections caused by *A. baumannii*.

## 2. Results

### 2.1. Gene Expression Profile Associated with Biofilm Formation

Following the alignment of clean reads with the reference genome of *A. baumannii* ATCC17978, a list of differentially expressed genes (DEGs) in the untreated and antibiotic-treated biofilm cells was identified (Appendix A). As shown in Table 1, 51.8% (1592/3075) of genes were differentially expressed in the untreated biofilm phase compared with their planktonic counterparts. In addition, 96 and 5 DEGs were classified as novel genes and sRNA, respectively. Of these 1592 genes, 20% (614/3075) were upregulated and 31.8% (978/3075) were downregulated in biofilm cells. For the imipenem-treated biofilm, 3.7% (106/2885) of the genes were differentially expressed. Seven and one DEGs were classified as novel genes and sRNA. Of the 106 genes, 45.3% (48/106) were upregulated and 54.7% (58/106) were downregulated in biofilm cells. For the colistin-treated biofilm, 12.6% (368/2912) of the genes were differentially expressed, 33 DEGs were classified as novel genes. Of the 368 genes, 64.9% (239/368) were upregulated and 35.1% (129/368) were downregulated in biofilm cells.

We classified the 1592 DEGs into different functional categories based on Gene Ontology (GO) annotation and Kyoto Encyclopedia of Genes and Genomes (KEGG) pathways enrichment. We focused on 211 genes that were linked to biofilms and antibiotic resistance for further analyses (Appendix A and Figure 1). To understand the effect of suboptimal doses of antibiotics, we subsequently identified 50 DEGs in biofilm cells upon exposure to a suboptimal dose of colistin or imipenem (Table 2).

### 2.2. Gene Expression Profile of Biofilm Treated with Sub-MIC of Imipenem

Table 1 summarizes the number of genes expressed in biofilm cells of *A. baumannii* ST1894 when treated with imipenem at sub-MIC. Of the 2885 total expressed genes, 3.7% (106) were differentially expressed in the imipenem-treated biofilm cells and the details in listed in Appendix A. Of these 106 DEGs, 78 were biofilm-specific, of which 48 were upregulated in sub-MIC of imipenem treated biofilm versus untreated biofilm cells. The upregulated genes with biofilm cells treated with sub-MIC of imipenem include genes encoding *pgaB*, genes encoding the fimbrial protein, AHL synthase, the T6SS protein ImpK, preprotein translocase subunit SecA, a cAMP-activated global transcriptional regulator and cyclic-AMP receptor protein (CRP), RND family drug transporter and sRNA00203, as shown in Table 2. Thirty genes were downregulated in treated versus untreated biofilm cells, including genes encoding the ATP-binding cassette (ABC) transporter, DNA gyrase, ribosome release factor (RRF), and 30S ribosomal protein S20 (Table 2).

### 2.3. Gene Expression Profile of Biofilm Treated with Sub-MIC of Colistin

Of the 2912 total expressed genes, 2.6% (368) were differentially expressed in the biofilm cells treated with Sub-MIC of colistin as illustrated in Appendix A. Of the 368 DEGs, 285 were biofilm-specific genes, of which 197 were upregulated and 88 were downregulated compared to untreated biofilm cells. We selected 44 DEGs in the biofilm cells that were either induced or repressed when treated with colistin, as shown in Table 2. Of the 44 DEGs, 28 and 16 were upregulated and downregulated, respectively, when the biofilm cells were treated with sub-MICs of colistin. The colistin-induced biofilm-specific genes include those genes encoding ABC-type iron transport proteins, fimbrial protein, lipoprotein (biofilm matrix), AHL synthase (AbaI), multidrug efflux pumps, OMP protein, and catalase and the *csu* operon. The biofilm-specific genes repressed by sub-optimal concentrations of colistin included genes encoding the LPS export system permease, DNA gyrase, RRF, 30S ribosomal protein S20, tRNA-i(6)A37 modification enzyme, and ATPase. This indicates that a sub-MIC of colistin can either activate or suppress biofilm-specific genes to promote the survival of biofilm cells in the presence of antibiotics.

### 2.4. Verification of Genes Induced or Repressed by the Sub-MICs of Imipenem and Colistin

To verify the RNA-seq results, 16 genes that were induced or suppressed by the sub-MICs of imipenem and colistin were selected for verification with qRT-PCR experiments (Figure 2 and Figure 3). These DEGs were associated with adherence, biofilm matrix synthesis, QS, β-lactam resistance, multidrug efflux pumps, replication and translation, environmental information processing, and ncRNAs (Figure 2 and Figure 3). A correlation coefficient of 0.98 was obtained from the linear regression plotted between the RT-qPCR and RNA-seq data, suggesting a strong positive correlation (Figure 2). The changes in gene expression measured using RT-qPCR displayed a pattern similar to the one seen in the RNA-seq data (*p* ≥ 0.05), suggesting that RNA-seq can be used interchangeably to describe transcriptional changes observed in biofilm and planktonic cells.

## 3. Discussion

The capacity of *A. baumannii* to form biofilms enhances its survival in adverse environments, making it a successful nosocomial pathogen [5,17,18,19]. In our previous study, we identified the non-MDR and strong biofilm-forming strain *A. baumannii* ST1894, the biofilm cells of which exhibited reversible antibiotic tolerance to colistin, imipenem, and ciprofloxacin [12]. This finding indicated that biofilms play a substantial role in the survival of *A. baumannii* by modifying its responses to antibiotics [12]. The phenotypic changes observed in this strain could be a result of alterations in gene expression in the biofilm cells and not irreversible genetic mutations. In this study, we analyzed the transcriptomic profiles between biofilm and planktonic cells, untreated biofilm cells and imipenem-treated or colistin-treated biofilms of *A. baumannii* ST1894. Upon conducting RNA-seq of the biofilm and biofilm cells treated sub-optimal doses antibiotics, we identified the genes involved in biofilm formation and biofilm specific antibiotic resistances.

In this study, an increase in the expression levels of the *pgaB* gene and genes encoding LPS biosynthesis components was observed in biofilm cells relative to planktonic cells. Other research groups have reported that deletion of the *pga* locus led to the loss of the strong biofilm formation phenotype of *A. baumannii*, demonstrating that this gene is essential for biofilm formation [20,21]. The production of EPS indicates that biofilm cells have reached the stage of irreversible adherence to the surface. The *pgaB* gene is involved in the biosynthesis of EPS, which is a significant component of the biofilm matrix [22].

Components of the biofilm matrix can limit the penetration of antimicrobial agents, such as bleach and antibiotics, into biofilm cells by binding to or consuming the antimicrobial agent [23,24]. However, when we observed antibiotic-treated biofilm cells using confocal imaging, we observed that most of the biofilm cells were killed by treatment with high concentrations of colistin, imipenem, and ciprofloxacin [12]. This observation suggests that the antibiotics can be passed through the matrix of the biofilm. Although the biofilm matrix might have reduced the penetration of antibiotics into the biofilm cells, this possibility needs to be experimentally verified.

We also observed the increased expression of *pgaB* and the gene coding for UDP-N-acetylglucosamine acyltransferase (novel00626) in biofilm cells exposed to sub-MICs of imipenem and colistin. This observation explains how sub-MICs of antimicrobials enhance biofilm formation which was described in a previous in-vitro study [16,25]. The biofilm matrix formation triggers stress-induced metabolic or transcriptional changes that increase resistance in cells exposed to sub-MICs of antibiotics [26,27].

We also observed that expression of the *pilA* gene, which encodes the fimbrial protein (A1S_3177) or type IV pilus assembly protein, was upregulated 222-fold in biofilm cells compared with planktonic cells. The expression of *pilA* also increased when the biofilm cells were treated with sub-MICs of imipenem and colistin.

A previous study reported that this type IV pilus assembly protein, commonly found in pathogenic *A. baumannii* strains, plays an essential role in host cell adhesion, biofilm formation, microcolony formation, and horizontal gene transfer [28,29]. In this study, the upregulated expression of type IV pili genes upon exposure of the cells to sub-MICs of imipenem and colistin might have activated signaling cascades associated with pathogenicity and antibiotic resistance in *A. baumannii*. When *A. baumannii* cells form biofilms in clinical settings, overexpression of the type IV pilus assembly protein upon exposure to sub-optimal concentrations of imipenem and colistin can increase the rates of conjugative gene transfer, thereby increasing the likelihood of developing irreversible antibiotic resistance and consequent therapeutic failure. Sub-MICs of antibiotics also promote mutation, leading to the emergence of antibiotic resistance [30,31]. This sequence of events can eventually result in the evolution of sensitive strains into resistant strains.

We observed that the *csu* operon (proteins CsuA, CsuC, and CsuE) was upregulated in biofilm cells. The *csu* operon encodes proteins involved in the chaperon-usher pili assembly mechanism, which is essential for the assembly of pili and the formation of biofilms [32]. The *csu* operon has been identified in pathogenic strains of *A. baumannii*, indicating its role as a virulence factor [33,34]. However, we observed that exposure to imipenem and colistin at sub-MICs did not significantly affect the expression of the *csu* operon.

QS is a regulatory mechanism that allows bacteria to communicate cell density information through the diffusion of small molecules and adjust their gene expression profiles accordingly [35,36]. All QS bacteria generate and release chemical signal molecules called autoinducers (AIs) that increase in concentration as a function of cell density [21,35,37]. AHLs are a class of AI signal used by Gram-negative bacteria to regulate various physiological processes such as conjugation, virulence factor production and biofilm formation [35,38,39,40,41,42].

AHLs have been identified as major components of biofilm formation in *A. baumannii* cells [43,44,45,46]. In this study, the genes encoding AHL synthase (E5A72_RS06870) were upregulated by 16.9-fold in biofilm cells compared with planktonic cells. The exposure of biofilm cells to sub-MICs of imipenem and colistin also induced genes encoding AHL synthase, which were upregulated by 12.2-fold and 5.8-fold, respectively. At a particular threshold, the binding of AHL to receptors within the cell promotes a signal transduction cascade that eventually changes the expression levels of specific genes involved in virulence and antibiotic resistance [42,45,47]. These changes in gene expression enable this pathogen to survive in adverse environmental conditions by promoting biofilm formation. When treated with sub-MICs of imipenem and colistin, the increased expression of genes encoding AHL synthase in biofilm cells suggests that a higher concentration of AIs is required to counteract adverse conditions encountered in the biofilm state.

Efflux pumps actively eliminate antimicrobial agents from intracellular targets, facilitating the reduced antibiotic susceptibility of biofilm and planktonic cells [46,48,49]. In the current study, we observed that 12 RND family MDR genes were differentially expressed in biofilm cells. Among these genes, novel00738 was upregulated in untreated biofilms and biofilm cells exposed to sub-MICs of imipenem. The product of this gene is functionally similar to the MexB-AcrB efflux pump, which confers resistance to β-lactams and cationic antimicrobial peptides. Poole et al. (1993) previously reported that deleting the genes encoding the MexAB-OprM efflux pathway in *Pseudomonas aeruginosa* resulted in hypersensitivity to many antimicrobial compounds [50,51,52,53,54].

The gene novel00738 encodes as a multidrug efflux pump that is active in β-lactam resistance pathways and is overexpressed when treated with sub-MICs of imipenem. The overexpression of novel00738 can result in tolerance to imipenem when biofilm cells are treated with high concentrations of β-lactam antibiotics. This finding is similar to one reported by He et al. [55], who demonstrated the role of the AdeFGH efflux pump in biofilm formation in response to low-concentration antimicrobial therapy. The increased expression of genes encoding efflux systems could reduce the cytoplasmic concentrations of bactericidal antibiotics to below the threshold required for antibacterial activity.

One of the main mechanisms underlying the development of antibiotic resistance is the pumping of antibiotics out of cells by efflux systems [55,56,57]. Efflux pumps can not only provide resistance to antibiotics used in clinical therapy but can also drive bacterial pathogenicity and persistence [58,59]. The increased expression of novel00738 in biofilm cells might be responsible for antibiotic tolerance and not resistance, as this strain was seen to revert to the susceptible form after treatment with a high concentration of antibiotics.

Analysis of the cationic antimicrobial resistance peptide path also revealed that novel00738 is functionally similar to *acrB.* The expression of *acrB* gene affects the expression of a group of efflux pump genes such as *tolC*, which is believed to confer tolerance to cationic peptides. The *tolC* gene is expressed at significantly higher levels in persister cells than in normal viable cells [59,60,61].

We also found that the *adeH* (acb: A1S 2306) gene was upregulated in biofilm cells and was further induced by treatment with sub-MICs of colistin. AdeH can pump colistin out of the cell and thereby confer tolerance to this antibiotic. Overexpression of AdeFGH has also been reported to facilitate the synthesis and transport of AHLs in *A. baumannii* during biofilm production [55,57,62]. We further observed a positive correlation between the expression of AIs and the upregulation of *adeH*, indicating that this gene might be involved in the transport of QS molecules, in addition to the expulsion of antibiotics.

We also observed that novel00626, which encodes UDP-N acetylglucosamine O-acyltransferase, appears to play a role similar to that of *lpxA*. This novel gene was highly upregulated in biofilm cells compared with their planktonic counterparts, and its expression level also significantly increased when biofilm cells were treated with sub-MICs of colistin. Previous studies have demonstrated that strains mutant for LOS production cannot survive desiccation, implying that the development of desiccation resistance is dependent on the composition of the outer membrane [4,63,64]. However, the processes that mediate desiccation resistance have not been studied and are currently being characterized. The upregulation of genes encoding UDP-N acetyl glucosamine O-acyltransferase during biofilm formation and in cells treated with sub-MICs of colistin is attributable to the role of LPS in the synthesis of the biofilm matrix.

We found that genes involved in DNA replication, transcription, and translation were differentially expressed in biofilm cells compared with planktonic cells. The genes involved in DNA replication were significantly downregulated in biofilm cells. The novel00171 gene, thought to encode DNA gyrase, was downregulated 32.4-fold in biofilm cells compared with planktonic cells. When treated with sub-MICs of imipenem, the expression of novel00171 decreased by a 42.2-fold relative to untreated biofilm cells. Such downregulation of DNA gyrase genes during biofilm formation has never been reported previously.

DNA gyrase is necessary for the replication and transcription of DNA. Reductions in intracellular gyrase proteins by >50% have been shown to affect cell growth [65,66]. The altered supercoiling of DNA due to gyrase depletion causes subsequent changes in the density of RNA polymerase in the transcription units, thereby altering transcription. The consequently reduced transcriptional activity generates a high number of slow-growing biofilm cells, which can also occur due to the limited availability of nutrients. The presence of slow-growing cells in biofilms may lead to the development of decreased susceptibility to antibiotics [67]. We also observed no substantial changes in the expression level of the novel00171 gene when the biofilm cells were exposed to sub-MICs of colistin.

The expression level of the gene encoding the RRF (*frr*) was reduced by 129.7-fold in biofilm cells. When the biofilm cells were treated with sub-MICs of imipenem and colistin, the expression level of *frr* decreased by 33.3-fold and 4.5-fold, respectively. The primary purpose of RRF is to recycle ribosomes for subsequent rounds of protein synthesis; it is thus essential for bacterial growth [68,69]. The downregulation of genes encoding RRFs in untreated biofilms and in antibiotic-treated biofilms might be due to the presence of slow-growing cells, which can confer antibiotic tolerance.

The expression of novel00490, which encodes the transcription termination factor Rho OS, was 113.7-fold higher in biofilm cells compared with planktonic cells. Expression of novel00490 increased significantly by 6.1-fold when the biofilm cells were treated with sub-MICs of imipenem. The transcriptional changes observed in both untreated biofilm cells and those exposed to sub-MICs of imipenem and colistin may shut down metabolic activity. Such a shutdown could directly inhibit other vital cellular activities and inactivate antibiotic targets.

We further observed that all of the DEGs involved in the citric acid cycle were downregulated, while all of those involved in glycolysis were upregulated. The downregulation of the citric acid cycle suggests that biofilm cells have lower metabolic rates than planktonic cells. This metabolic quiescence might contribute to the reduced susceptibility to antibiotics observed in hyper biofilm-forming strains such as *A. baumannii* ST1894. It also implies that the persisters observed in the biofilms of *A. baumannii* ST1894 after treatment with high concentrations of bactericidal antibiotics might have emerged a result of reduced metabolic activity.

The transcriptomic analysis revealed that genes encoding acinetobactin biosynthesis proteins were downregulated in biofilm cells relative to their planktonic counterparts. The identified genes encode iron-induced proteins, such as the iron storage protein Bfr, metabolic proteins, such as AcnA, AcnB, GlyA, SdhA and SodB, and lipid biosynthesis proteins [70]. The reduced expression of these genes can further downregulate the expression of genes involved in aerobic respiration. Similar patterns were observed in our current study: the downregulation of genes involved in the citric acid cycle suggests slower metabolic rates in biofilm cells. This reduced metabolic activity could further increase the antibiotic tolerance of biofilm cells compared with planktonic cells.

In addition to the novel protein-coding transcripts, the total RNA-seq data showed that 5 of 138 sRNAs were differentially expressed in biofilm cells relative to planktonic cells. Of these sRNAs, sRNA00203 exhibited the highest expression levels in biofilm cells that were untreated or treated with sub-MICs of imipenem. sRNAs regulate protein expression by complementing target mRNAs and interacting with mRNA transcripts at or near the RBS [71,72,73]. sRNA00203, an ncRNA, is believed to play roles in various cellular processes, including the regulation of gene expression. sRNAs are genetic regulators that enable biofilm cells to recognize environmental signals and relay information in the form of metabolic changes with significant physiological effects during biofilm formation [45,74,75].

Overall, we observed that the hyper biofilm-producing strain *A. baumannii* ST1894 demonstrated a high degree of reduced susceptibility to antibiotics. The capacity of *A. baumannii* ST1894 to survive the effects of bactericidal antibiotic exposure during biofilm formation might emanate from genetic changes that arise due to this exposure in the biofilm state.

This is the first study to characterize transcriptional changes in biofilm cells in response to treatment with sub-MICs of imipenem and colistin. Several novel findings of this study are reported here. First, the consistent upregulation of genes involved in biofilm matrix synthesis (*pgaB*), multidrug efflux pump (novel00738) and LPS synthesis (novel00626) in *A. baumannii* in response to treatment with sub-MICs (half of the MIC) of imipenem and colistin may lead to increased biofilm production. This finding illustrates the possible relationship between low-dose antimicrobial therapy and enhanced biofilm production, which can occur during *Acinetobacter* infections. Second, this study showed a reduced expression of genes linked to acinetobactin biosynthesis and protein synthesis (novel00171, RRF, novel00584) during biofilm formation, which might slow down metabolism. Such changes enhance biofilm-specific antibiotic resistance and environmental resilience when the cells are exposed to antibiotics. Third, an upregulation of sRNA00203 in the biofilm cells was observed. As sRNAs are involved in the regulation of many cellular processes, activation of sRNA gene expression may in turn affect the expression of biofilm-specific antibiotic resistance. Based on our findings of the transcriptome study, multiple genetic factors account for the decreased susceptibility of biofilm cells to antibiotics, the precise interactions between various factors warrant further investigation.

## 4. Materials and Methods

### 4.1. Bacterial Strain and Culture Conditions

The *A. baumannii* strain used for comparative transcriptome analysis was *A. baumannii* ST1894, which was a clinical strain isolated from the sputum of a patient suffering from a lower respiratory tract infection. *A. baumannii* ST1894 has been characterized in our previous investigation and shown to be a hyper biofilm-former and was susceptible to colistin and imipenem when grown in the planktonic phase. However, the strain exhibited a high degree of resistance to colistin and imipenem when grown in the biofilm phase [12]. The strain was grown in LB broth at 37 °C with shaking (180 rpm) and stored at −80 °C in LB broth containing 20% glycerol until being used.

### 4.2. Biofilm Preparation and Exposure to Antibiotics

A single colony was picked from a pure culture grown on LB agar plates and then inoculated into 20 mL of LB broth to obtain a planktonic culture. Similarly, five colonies were selected and inoculated into a larger Petri dish (150 mm diameter) containing 100 mL of LB broth. The planktonic and biofilm cultures were incubated at 37 °C for 48 h [76]. After 47 h of incubation, 62.5 mg/mL of imipenem (half of the MIC) and 250 mg/mL of colistin (half the MIC) were added to the biofilm incubated at 37 °C for another hour. After 48 h of cultivation, all of the single cells were carefully washed three times with maximum recovery diluent (1 g peptone and 9 g of NaCl per liter of distilled water) without disturbing the biofilm. 

Subsequently, the biofilm cells attached to the Petri dish surface were removed using a plastic cell scraper and then resuspended in maximum recovery diluent. The cells were placed on ice and then washed thrice with 1 mL of cold phosphate-buffered saline (PBS). After washing, the biofilm and planktonic cultures were centrifuged for 15 min at 3500× *g* and 4 °C. The pellets obtained from the biofilm cells, antibiotic-treated biofilm cells, and planktonic cells were stored at −80 °C until they were processed for RNA extraction.

### 4.3. RNA Extraction

Cell pellets prepared from the planktonic, biofilm, imipenem-treated (half of the MIC), and colistin-treated biofilm cells (half of the MIC) were processed for total RNA isolation using a PureLink RNA Mini Kit (Ambion, Life Science Technologies, Carlsbad, CA, USA). The isolated RNA was treated with DNase to eliminate contaminating genomic DNA using TURBO DNase (Life Science Technologies).

### 4.4. RNA Sequencing

Library preparation and RNA-seq were performed by Groken Bioscience Ltd, Hong Kong. Three micrograms of input RNA were used to construct the libraries using the NEBNext UltraTM Directional RNA Library Prep Kit for Illumina (NEB, Northborough, MA, USA). The libraries were sequenced on an Illumina HiSeq2500 platform (San Diego, CA, USA) with 150PE reads.

### 4.5. Bioinformatics Analysis

The clean reads were obtained by quality filtering and trimming of adapters by using custom Perl scripts and Trimmomatic v0.3032 (Cambridge, UK) with the default parameters.

The reference genome of *A. baumannii* ATCC17978 with RefSeq assembly accession GCF_004797155.2 (latest) was retrieved from the National Center for Biotechnology Information (NCBI) website. The reads were mapped to the reference genome using Bowtie 2 v2.2.3 with default parameters [77]. HTSeq v0.6.1 was used to count the number of reads mapped to each gene, and the fragments per kilobase of transcript sequence per million base pairs (FPKM) were calculated to quantify the level of gene expression. Simultaneously, the log_2_ (FPKM biofilm/FPKM planktonic), which accounts for the effects of sequencing depth and gene length on the read count, was determined [78]. The results show the number of genes with different expression levels and the expression levels of single genes. An FPKM value of 0.1 or 1 was set as the criterion for determining the expression levels of the target groups. The raw sequences were deposited in the NCBI Sequence Read Archive (SRA) under the BioProject ID PRJNA892543 and BioSample accession numbers SAMN31387122, SAMN31387123, SAMN31387124, SAMN31387125.

The sequencing reads were annotated using the R Bioconductor package [79]. Normalization of gene expression was performed using the edge program package and a single scaling factor. The DEGseq R package (1.18.0) was used to estimate the DEGs between biofilm and planktonic cells, untreated biofilms, imipenem-treated biofilm cells, and untreated biofilms and colistin-treated biofilm cells. The fold change and level of significance, which indicate differential expression, were evaluated using a model based on the binomial distribution (which could be approximated by a Poisson distribution) [80]. Genes with an expression level of log_2_ fold change > 1 and a corrected *p* (*q*) ≤ 0.005 were considered to be differentially expressed. Genes were considered to be differentially expressed if they fulfilled the following criteria: (1) the biofilm cells had a normalized gene expression value > 2-fold that of the planktonic cells; and (2) the antibiotic-treated biofilm cells had a normalized expression value > 2-fold that of untreated biofilm cells. The DEGs were then used for Gene Ontology (GO) and Kyoto Encyclopedia of Genes and Genomes (KEGG) enrichment pathway analyses.

#### 4.5.1. Novel Gene and Gene Structure Analysis

The RNA-seq reads were mapped to the reference genome of *A. baumannii* ATCC17978 using Rockhopper, and the novel genes were compared to known gene structures [81]. The novel transcripts were BLASTx (cut-off: e value < 1 × 10^−5^) against non-redundant protein database. Novel transcripts with NR protein sequence annotations were considered to be novel potential protein-coding transcripts. The transcription start sites (TSSs) and termination sites of operons were predicted based on the positions of the reads in the reference genome using Rockhopper. A 700-bp sequence in the upstream TSS was extracted and used to identify the promoter according to the time-delay neural network (TDNN) method.

#### 4.5.2. The GO and KEGG Enrichment Analyses

The GO seq R package was used to perform the GO enrichment study of the DEGs; GO terms with an adjusted *p* < 0.05 were considered to be enriched [82]. KEGG enrichment analysis was conducted to identify significantly enriched metabolic pathways or signal transduction pathways across antibiotic-treated biofilms, untreated biofilms, and planktonic cells. KOBAS (Dalian, China) was used to evaluate the statistical enrichment of DEGs in KEGG pathways. KEGG pathways with an adjusted *p* < 0.05 were considered to be significantly enriched in DEGs.

### 4.6. Verification of Genes of Induced or Repressed by Exposure to Sub-MICs of Imipenem and Colistin

RT-qPCR was used to verify the differentially expressed genes obtained from the RNA-seq data [81,83]. Sixteen genes associated with biological functions such as matrix formation, QS, β-lactam resistance, cationic antimicrobial peptide resistance, bacterial secretion system, and the two-component system were selected for RT-qPCR verification (Table 3). The same RNA samples (technical replicates) used for transcriptomic analysis were reversely transcribed to cDNA followed by RT-qPCR. In addition, RNA samples prepared from two more independent experiments of antibiotic exposure and prepared under the same biological conditions were used as biological replicas. Thus, for each group (planktonic, untreated biofilm, imipenem-treated or colistin-treated biofilm), three replica RNA samples were subjected to RT-qPCR.

The sequences of the 16 selected genes were retrieved from the *A. baumannii* ST1894 transcriptome data and used as references for the design of primers and probes for qPCR. The primers and probes shown in Table 4 were designed using Primer3 plus (Boston, USA). A two-step protocol was used to perform RT-qPCR, 500 ng of pure RNA samples were reverse-transcribed using a Luna Script RT Supermix Kit (NEB, Massachusetts, USA). The prepared cDNA and no-RT control reactions were diluted at 1:100 in nuclease-free water to be used for RT-qPCR reactions. The SYBR Green RT-qPCR reaction was prepared as a 20-µL mixture by adding 10 µL of Luna Universal qPCR Master mix (NEB, Massachusetts, USA), 0.5 µL of forward primer (10 µM), 0.5 µL of reverse primer (10 µM), 2 µL of 1:10 diluted cDNA and 7 µL of nuclease-free water. A no-RT control was run in parallel.

The SYBR Green qPCRs were performed on a ViiA 7 Real-Time PCR system (Applied Biosystems, USA) using the following parameters: 1 min at 95 °C; 40 cycles of 95 °C for 15 s and 60 °C for 30 s; and 1 min at 72 °C. A melting curve analysis was added to ensure the specificity of the PCR product.

TaqMan assays were performed for the novel transcripts and sRNAs using the Luna Universal Probe qPCR Master mix (NEB). The reaction parameters set on the ViiA 7 Real-Time PCR were: 1 min at 95 °C and 40 cycles of 95 °C for 15 s and 60 °C for 30 s. For the SYBR Green and TaqMan qPCR experiments, no-RT controls were used for each target gene. The expression levels of three housekeeping genes (*rpoD*, *gyrB,* and E5A72 RS18355) were evaluated across all of the RNA-seq data. *gyrB* displayed the least variability across different sample groups, and therefore, the cycle threshold (C_T_) values of all 16 target genes were normalized using *gyrB* as an internal control.

## 5. Conclusions

In this investigation, we demonstrated that exposure of *A. baumannii* ST1894 to suboptimal doses of imipenem and colistin increased the expression of genes involved in biofilm formation and antibiotic resistance while decrease the expression of genes involved in protein synthesis in the biofilm state. This confirmed our previous observation that exposure of *A. baumannii* ST1894 biofilm to suboptimal antibiotic doses induced biofilm formation and antibiotic resistance. We also showed that expression of non-coding sRNA00203 was highly induced in either untreated or imipenem-treated biofilm cells. As sRNAs play critical regulatory roles in many cellular processes, this adds to the complexity and versatility of biofilm regulatory mechanisms and confers a survival advantage to the pathogen.

## Figures and Tables

**Figure 1 ijms-23-12705-f001:**
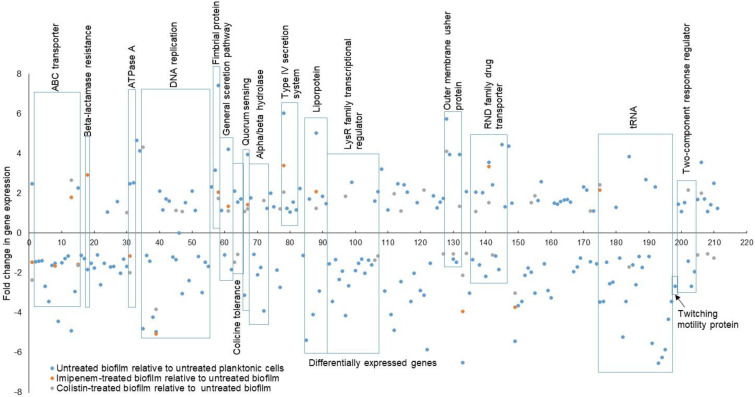
Distribution of significantly up- or downregulated genes belonging to functional categories associated with virulence and antibiotic resistance in biofilm. The blue dots represent the DEGs between untreated biofilm and planktonic cells. The orange dots represent the DEGs between untreated and imipenem-treated biofilm cells. The grey dots represent the DEGs between untreated and colistin-treated biofilm cells.

**Figure 2 ijms-23-12705-f002:**
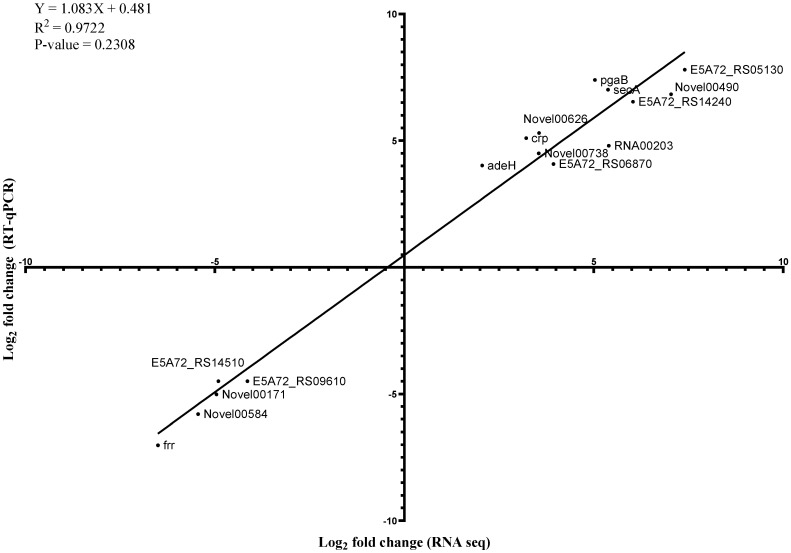
Relationship between gene expression fold changes obtained from RNA sequencing and RT-qPCR. A total of 16 genes associated with virulence and antibiotic resistance in biofilm were compared. The measured log_2_ fold change in gene expression of biofilm cells relative to planktonic cells are plotted against the RNA sequencing data (statistical goodness of fit value provided).

**Figure 3 ijms-23-12705-f003:**
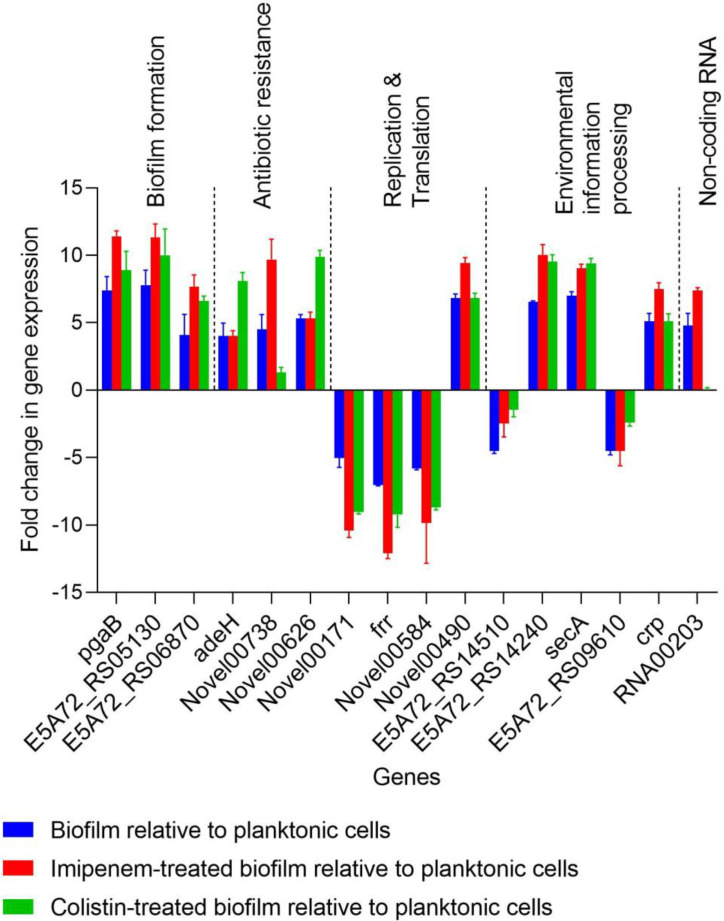
Effect of imipenem and colistin at sub-MICs on genes associated with virulence and antibiotic resistance in *A. baumannii* ST1894 biofilm. *pgaB*: biofilm matrix synthesis; E5A72_RS05130: fimbrial protein; E5A72_RS06870: quorum sensing; *adeH*: multidrug efflux pump; Novel00738: resistance–nodulation–cell division (RND) efflux pumps; Novel00626: peptidoglycan biosynthesis; Novel00171: DNA replication; *frr*, Novel00584, Novel00490: translation; E5A72_RS14510: ATP-binding cassette (ABC) transporter; E5A72_RS14240, *secA*: bacterial secretion system; E5A72_RS09610, *crp*: two-component system; RNA00203: non-coding RNA.

**Table 1 ijms-23-12705-t001:** Summary of genes differentially expressed in untreated biofilm and biofilm exposed to imipenem and colistin at sub-MIC dosages.

Group	Transcribed Genes	Differentially Expressed			
		Total No. (%)	Novel Genes	Genes Encoding Hypothetical Proteins	Small RNA
Biofilm vs. planktonic cells	3075	1592 (51.8)	96	431	5
Imipenem-treated biofilm vs. untreated biofilm cells	2885	106 (3.7)	7	11	1
Colistin-treated biofilm vs. untreated biofilm cells	2912	368 (12.6)	33	74	0

**Table 2 ijms-23-12705-t002:** Biofilm-specific genes induced or repressed by sub-MICs of imipenem and colistin.

Gene_id	KEGG_ID	Fold Change in Gene Expression			Function/Product
		Untreated Biofilm Relative to Untreated Planktonic	Imipenem-Treated Biofilm Relative to Untreated Biofilm	Colistin-Treated Biofilm Relative to Untreated Biofilm	
Lipoprotein					
E5A72_RS12100	acb: A1S_0938	5.0363	2.09	1.23	Lipoprotein (biofilm matrix)
ATP-binding cassette (ABC) transporter					
E5A72_RS02170	acb: A1S_2611	2.3183	2.14368	1.02	Transport protein of outer membrane lipoproteins
E5A72_RS16335	acb: A1S_1722	−1.5054	−1.639	0.095	ABC transporter ATP-binding protein
E5A72_RS14510	acb: A1S_1359	−4.9058	1.7825	2.6702	ABC-type Fe^3+^ transport system
E5A72_RS08785	acb: A1S_0229	2.0395	1.5784	−1.6349	Lipopolysaccharide export system permease protein
E5A72_RS15155	acb: A1S_1482	−3.3293	1.0455	2.0257	D-and L-methionine transport protein
β-lactam resistance					
E5A72_RS00940	acb: A1S_2367	−1.8362	2.92	1.16 *	*ampC*; β-lactamase
Novel00738	acb: A1S_2736	3.5549	3.3532	−1.013	Resistance–nodulation–cell division (RND) family drug transporter
DNA replication and repair					
Novel00171	acb: A1S_2626	−4.9582	−5.0584	−3.8428	DNA gyrase
E5A72_RS14680	acb: A1S_1389	−1.3449	−0.0409 *	1.1446	DNA polymerase V component
E5A72_RS03815	acb: A1S_2918	−3.0414	−0.08 *	1.0746	DNA repair protein
Fimbrial protein and csu operon					
E5A72_RS05130	acb: A1S_3177	7.4033	2.0455	1.7527	Fimbrial protein
E5A72_RS19200	acb: A1S_2217	5.7238	NA	2.1677	Protein CsuA
E5A72_RS19190	acb: A1S_2215	1.3111	−0.683 *	1.0371	Protein CsuC
E5A72_RS19180	acb: A1S_2213	3.9517	0.2679 *	4.1059	Protein CsuE
Bacterial secretion system					
E5A72_RS14240	acb: A1S_1310	6.0356	3.387	2.045	Type VI secretion system protein
E5A72_RS15605	acb: A1S_1564	4.2	1.3587	1.1006	General secretion pathway protein J
E5A72_RS03500	acb: A1S_2862	5.376	1.65783	2.349	Preprotein translocase subunit SecA
Two-component regulatory system					
E5A72_RS09610	acb: A1S_0399	−4.1397	−0.346 *	1.8361	LysR family transcriptional regulator
E5A72_RS12380	acb: A1S_0992	1.617	−0.132 *	−1.3517	
E5A72_RS13515	acb: A1S_1182	3.216	2.06	0.065 *	cAMP-activated global transcriptional regulator CRP
Novel00822	acb: A1S_0685	−1.4033	1.788 *	2.4156	Two-component response regulator
Quorum sensing or quenching					
E5A72_RS06870	acb: A1S_0109	3.9355	1.43253	1.21081	Acyl homoserine lactone (AHL) synthase (*AbaI*)
E5A72_RS16835	acb: A1S_1809	−3.9015	0.267 *	1.6271	Hydrolase transmembrane protein
Multidrug efflux pump					
E5A72_RS00640	acb: A1S_2306	2.0527	−0.083 *	2.1244	RND efflux transporter
E5A72_RS11940	acb: A1S_0908	2.4322	−0.4375	1.0937	RND family multidrug resistance secretion protein
Transcription and translation					
E5A72_RS17735	acb: A1S_1974	−6.5036	−3.9244	−2.1244	Ribosome releasing factor
Novel00584	acb: A1S_1617	−5.4437	−3.7365	−3.01	30S ribosomal protein S20
Novel00490	-	7.0401	1.1587	0.415	Transcription termination factor Rho OS
E5A72_RS04400	acb: A1S_3029	−3.4737	2.161	2.4156	tRNA-Arg
Peptidoglycan biosynthesis					
E5A72_RS10790	acb: A1S_1965	1.6823	−0.1303 *	−1.6917	UDP-N-acetylglucosamine 1-carboxyvinyltransferase
Novel00626	acb: A1S_1965	3.5549	NA	2.1677	UDP-N-acetylglucosamine acyltransferase
E5A72_RS05230	acb: A1S_3203	1.0844	−0217*	−1.032	UDP-N-acetylmuramoylalanyl-D-glutamate-2, 6-diaminopimelate ligase
E5A72_RS17730	acb: A1S_1973	2.513	−0.0783 *	−1.2536	Undecaprenyl pyrophosphate synthetase
E5A72_RS07480	acb: A1S_2968	−2.7334	0.4605 *	1.2183	Hypothetical protein
E5A72_RS05220	acb: A1S_3200	1.7367	0.0003 *	−1.0371	Phospho-N-acetylmuramoyl-pentapeptide transferase
Outer membrane protein					
E5A72_RS11985	-	−4.8714	−0.1244 *	1.9751	OprD family outer membrane porin
E5A72_RS18480	acb: A1S_2076	−2.4293	0.4473 *	1.1074	Outer membrane porin receptor for Fe (III)- coprogen
E5A72_RS01795	acb: A1S_2538	−3.1257	1.4605 *	2.1677	Outer membrane protein CarO precursor
Transcriptional regulators and others					
E5A72_RS13590	acb: A1S_1197	−2.5865	−1.2664	0.240 *	Extracellular nuclease
E5A72_RS03015	acb: A1S_2767	−1.6789	0.7044	1.0387	AraC family transcriptional regulator
E5A72_RS04330	acb: A1S_3229	3.8412	0.25977	−1.6917	tRNA-i(6)A37 modification enzyme
E5A72_RS01785	acb: A1S_2536	2.4782	−1.1416	−1.9789	ATPase
E5A72_RS14655	acb: A1S_1386	−4.8062	NA	4.3282	Catalase; K03781 catalase
E5A72_RS01830	acb: A1S_2546	−1.9701	0.9841	1.5253	Secreted trypsin-like serine protease
E5A72_RS05375	acb: A1S_3227	2.0827	−0.02825	−1.013	RNA binding protein
E5A72_RS16050	acb: A1S_1670	−3.0262	0.598 *	−3.01	Secretion protein HlyD
E5A72_RS02070	acb: A1S_2592	1.5561	0.0036	−1.0789	Group A colicins tolerance protein
E5A72_RS02065	acb: A1S_2591	2.0955	−0.0461	−1.4623	Group A colicins tolerance protein
Non-coding RNA					
Gene encoding sRNA00203	-	5.3951	2.38935	−0.389 *	

Adjusted *p* (pad) < 0.004; * pad > 0.005; NA: not applicable. Minus sign (−): downregulated gene.

**Table 3 ijms-23-12705-t003:** Lists of genes induced or repressed by exposure to sub-MICs of imipenem and colistin and selected for RT-qPCR verification.

mRNA_ID	KEGG_ID	KEGG Annotation	Strand	Start	End	Length (bp)	Pathway
E5A72_RS12100	acb: A1S_0938	*PgaB*	+	2520896	2522890	1995	Biofilm matrix
E5A72_RS05130	acb: A1S_3177	Fimbrial protein	−	1078917	1079381	2538	Two-component system
E5A72_RS06870	acb: A1S_0109	Homoserine lactone synthase	+	1427610	1428176	567	Quorum sensing
E5A72_RS00640	acb: A1S_2306	RND^2^ efflux transporter	+	116590	118041	555	Multidrug efflux system
Novel00738	acb: A1S_2736	RND family drug transporter	+	617454	623989	6536	β-lactam resistance
Novel00626	acb: A1S_1965	UDP-N acetylglucosamine acyltransferase	−	3676375	3677683	1309	Cationic antimicrobial peptide resistance
Novel00171	acb: A1S_2626	DNA gyrase	−	483250	484821	1572	DNA replication
E5A72_RS17735	acb: A1S_1974	Ribosome releasing factor	−	3685962	3686516	555	Translation
Novel00584	acb: A1S_1617	30S ribosomal protein S20	−	3284582	3284881	300	
Novel00490	-	Transcription termination factor Rho OS	−	2214580	2215974	1395	Transcription
E5A72_RS14510	acb: A1S_1359	ABC^3^-type Fe^3+^ transport system	+	3025424	3026461	1038	ABC transporters
E5A72_RS14240	acb: A1S_1310	K11892 type VI secretion system protein ImpK	+	2971603	2972409	807	Bacterial secretion system
E5A72_RS03500	acb: A1S_2862	Preprotein translocase subunit SecA	−	756066	758789	2724	
E5A72_RS09610	acb: A1S_0399	LysR family transcriptional regulator	+	1993774	1994670	897	Two-component system
E5A72_RS13515	acb: A1S_1182	CRP^4^ transcriptional regulator	−	2818333	2819040	708	
sRNA00203	-	-	−	1245743	1245795	53	Non-coding RNA

**Table 4 ijms-23-12705-t004:** Primers and probes used for verification of expression levels of the 16 selected DEGs.

Gene ID	Amplicon Size (bp)	Primer/Probe	Sequence (5′ to 3′)
E5A72_RS12100 (*pgaB*)	105	E5A72_RS12100-F	CGGATGCGAATGGTTCTGC
		E5A72_RS12100-R	GCGTACGGGTTTGAATTTGC
E5A72_RS05130	217	E5A72_RS05130_4_F	CCGAAGGTACAGCTAACAGTG
		E5A72_RS05130_4_R	CCACCCACATTTGCATTTACT
E5A72_RS06870	121	E5A72_RS06870_F	GCCAGACTACTACCCACCAC
		E5A72_RS06870_R	CTACGGCTGAAAACCTTGAT
E5A72_RS00640	108	E5A72_RS00640_F	TCAGGCTTCACGTGCACTAC
		E5A72_RS00640_R	AAACCGAGTGAAGCTGGAGA
Novel00738	79	Novel00738_F	GCTGCCATTACTCGTTTACCT
		Novel00738_R	CAGGACGGCTCTCAACAAC
		Novel00738_IN	FAM-GGCAAGCTGTAGCGATGCTTGTTAAT-TAMRA
Novel00626	110	Novel00626_F	CGCATCGTTACCCATTCTT
		Novel00626_R	GAAATGCCCTTGTAGGAACTCT
		Novel00626_IN	FAM-TTGGTTGATCGTGTGACTGAAGTTACTGA-TAMRA
Novel00171	109	Novel00171_F	CATTGCCGGATGTGAGAG
		Novel00171_R	ACACGAGCAGATTTCTTGTAGG
E5A72_RS17735 (*Frr*)	98	E5A72_RS17735_F	GCGAAAGTTGCTATCCGTAA
		E5A72_RS17735_R	GCACGACGCTCATCATCT
Novel00584	114	Novel00584_F	TGCGTTCTATGGTTCGTACTT
		Novel00584_R	GCACGACGCTCATCATCT
Novel00490	109	Novel00490_F	TTAGCCCGTGCATACAACAC
		Novel00490_R	TAGCCCGTGCATACAACAC
		Novel00490_IN	FAM-TGGTGTGGATGCACATGCTTTAGAAC-TAMRA
E5A72_RS14510	103	E5A72_RS14510_F	AGGTTTAGGCTGGGAAATGG
		E5A72_RS14510_R	ATTTGCTGCTTTGCTTACCG
E5A72_RS14240	110	E5A72_RS14240_1_F	GCACGAGTAGGCGATGAA
		E5A72_RS14240_1_R	AAAGGTAGCTCACGATGGATAA
E5A72_RS03500 (*secA*)	107	E5A72_RS03500_F	GACATTATTGCTCAGGCAGGT
		E5A72_RS03500_R	GCAAGTTTCGCTTTCCAGTT
E5A72_RS09610	86	E5A72_RS09610_3_F	AAGGTGGAACTGTGATGATGG
		E5A72_RS09610_3_R	AATTCCCAAACCTGCACAAG
E5A72_RS13515	118	E5A72_RS13515_3_F	ATCGACCTATCTTCACAACCAG
		E5A72_RS13515_3_R	ATACACGGCCAACCATTTC
sRNA00203	76	sRNA00203_4_F	GCATAAAAACCTCTTGAAACTGTTC
		sRNA00203_4_R	AGCGTTCATTTCAACCGATA
		sRNA00203_4_IN	TCAAGTTCCTTATGATCTCTTCCTTGA
*gyrB*	93	gyrB-F	ACGATTTACCGTGCTGGTC
		gyrB-R	GGTATTATCCGTTTCACCAATC
		gyrB_IN	FAM-TATCATCATGGTGATCCGCAATATCC-TAMRA

## Data Availability

The raw RNA sequencing datasets used for this study were deposited at the NCBI Sequence Read Archive; https://submit.ncbi.nlm.nih.gov/subs/sra under Bioproject ID number PRJNA892543; BioSample accession numbers SAMN31387122, SAMN31387123, SAMN31387124, SAMN31387125.

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
