# Peer review of "Effects of Sub-Minimum Inhibitory Concentrations of Imipenem and Colistin on Expression of Biofilm-Specific Antibiotic Resistance and Virulence Genes in Acinetobacter baumannii Sequence Type 1894"

_ijms, 2022, doi:10.3390/ijms232012705_

Round 1

Reviewer 1 Report

General comments: The work is very interesting. However, results presentation is not well performed and tables and figures are not well presented or mentioned in the text. Please correct all missing points and careful review the construction of the manuscript.

Specific comments:

·         The last paragraph of the introduction, Lines 74-79, you show results, which is not correct. In this paragraph you should only include the goals of the work and not results.

·         In line 83 please write the full name of the abbreviation DEGs. Other abbreviatures should be clarified before appear in the text.

·         Something is missing in Lines 85-93, I do not understand the meaning of all sentences.

·         Table 1 appears without being mentioned in the text. Possibly this is the table S1? Please correct this point.

·         Lines 104, 107,132, 135, 141, 142, 158, 162, 487 there are several errors in references.

·         Figure 1 is not well presented. Please correct all the manuscript presentation.

·         Once again. Table 2 is not mentioned in the text. Also, it is not possible to understand if it was imipenem or colistin used, or if the genes where up or downregulated with this table. Please re-build this table. All formulas must be explained.

·         Line 121 the sentence is not finished.

·         Figure 2 is in the middle of the text, please correct that.

·         It is necessary to include references in the manuscript that supports the sentences in lines 191, 193, 243, 244, 279, 321, 323, 463, 473, 484.

·         Please write “in vitro” in italics

·         In the line 409 when you mention “biofilm culture” include a reference to the model used.

·         Table 3 and 4 appears without being mentioned in the text.

·         Line 501 and 502 is missing information.

Author Response

Dear Reviewer

Greeting 

I replied to the suggested comments as indicated in the attached documents !

with best regard

Reviewer 2 Report

The authors reported the analysis of gene expression in Acinetobacter baumannii treated with sub-MIC of colistin and imipenem and will serve as a useful reference for the understanding biofilm formation under antibiotics condition. However, there are some points which should be modified in the manuscript.

 1. There are several areas where file merging is not working. It is difficult to read the manuscript. Especially, Figure 2 is repeated.

2. Figure 2 image is unclear.

3. "Errors" are displayed in the manuscript.

4. Which is supplemental Table S2?

5. Discussion section is a little bit long. Only the results of reference 11 are mentioned in a few paragraphs. Please discuss simple more.

6. In Materials and Methods, bacteria strains should be explained simply more. In addition, MICs should be shown in the text.

Author Response

Dear reviewer 

Greeting

we replied to the suggested comments as indicated in attached document! we appreciate your constructive feedback !

with best regards

Abebe

Reviewer 3 Report

The manuscript needs significant editing as there are blank spaces throughout the manuscript. The figures shows error message :Reference source. ot found and are of low quality. There are blank pages within the manuscript. The authors should rewrite and format the manuscript as per the journals guidelines and resubmit it. 

Author Response

Dear Reviewer

Greeting

we replied to suggested the comments as indicated in attached document ! We appreciate your constructive feedback!

with best regards

Round 2

Reviewer 1 Report

The authors have performed all the corrections suggested.

Reviewer 2 Report

Based on the reviewers' comments, the authors revised the manuscript.

This is acceptable.

Reviewer 3 Report

The authors have made significant changes to the manuscript. Overall, the work is solid and materials and methods are clearly written. The figure now look much better.